# Using a health equity lens to measure patient experiences of care in diverse health care settings

Annette J. Browne[1]☺*, Colleen Varcoe[1]☺, Marilyn Ford-Gilboe[2]☺, C. Nadine Wathen[2]☺, Erin Wilson[3‡], Vicky Bungay[1‡], Nancy Perrin[4]☺

1 School of Nursing, Faculty of Applied Science, The University of British Columbia, Vancouver, BC, Canada, 2 Arthur Labatt Family School of Nursing, Western University, London, ON, Canada, 3 School of Nursing, University of Northern British Columbia, Prince George, BC, Canada, 4 Johns Hopkins University School of Nursing, Baltimore, MD, United States of America

☺ These authors contributed equally to this work.
‡ EW and VB also contributed equally to this work.
* Annette.browne@ubc.ca

**Data Availability Statement:** Data cannot be shared publicly because of the datasets generated and/or analyzed for this study are not publicly available due to issues of safety, sensitivity and

## Abstract

People who are structurally disadvantaged and marginalized often report poor health care experiences, such as inequitable treatment, due to intersecting forms of stigma and discrimination. There are many measures of patient experiences of care, however, few are designed to measure equity-oriented health care. In alignment with ongoing calls to integrate actions in support of health equity, we report on the development and testing of patient-reported experience measures that explicitly use a health equity and intersectional lens. Our analysis focuses on two different scales: the *Equity-Oriented Health Care Scale—Ongoing*, which was evaluated in primary health care settings where patients have an ongoing relationship with providers over time, and the *Equity-Oriented Health Care Scale—Episodic*, which was tested in an emergency department where care is provided on an episodic basis. Item Response Theory was used to develop and refine the scales. The psychometric properties of each scale were also evaluated. The *Equity-Oriented Health Care Scale—Ongoing* was first tested with a cohort of 567 patients. The *Equity-Oriented Health Care Scale—Episodic* was subsequently tested in an emergency department setting with 284 patients. Results of the Item Response Theory analysis for each scale yielded a brief index that captured the level of equity-oriented care when care is ongoing (12 items) or episodic (9 items). Both scales showed evidence of internal consistency and concurrent validity, based on a high correlation with quality of care. They are brief, easy-to-administer patient-reported experience measures that can support organizations to monitor quality of care. Their availability enhances the possibility of measuring equity-oriented health care in diverse contexts and can provide nuanced understandings of quality of care through an intersectional and equity lens.

privacy. Participants in this study consented only to sharing of their research data within the immediate team and not with external parties. Public deposition would breach compliance with the protocols approved by our research ethics boards. Data may be available from the UBC Behavioral Research Ethics Board (contact via wendy. bond@ubc.ca) for researchers who meet the criteria for access to confidential data.

**Funding:** This study is funded by the Canadian Institutes of Health Research (grant #PJT148832), which was awarded to C.V. and A.B.. N.W. was supported by a SSHRC (Tier 1) Canada Research Chair. M.F.G. was supported by the Women's Health Research Chair in Rural Health. The funding body had no role in study design, collection, analysis, and interpretation of data or in writing this manuscript.

**Competing interests:** The authors have declared that no competing interests exist.

## Introduction

A growing body of evidence continues to demonstrate that greater equity in health and health care is associated with improved population health [1–3]. Achieving aims related to health equity requires focusing on people who have the least access to the social determinants of health and face the greatest barriers to health care, including those who are most structurally disadvantaged in our societies. The idea of structural disadvantage recognizes that inequities are structural because they are embedded in social, economic, and health care policies and practices, and contribute to tangible, negative impacts on health, quality of life, and well-being. For example, people who are structurally disadvantaged experience poorer outcomes on many measures of health and report poorer health care experiences [4–7]. Research also shows that people's health care experiences influence their timely access to health care and their overall health. Importantly, research continues to confirm that people who have negative health care experiences or who anticipate such experiences, including experiences arising from stigma and discrimination, are deterred from accessing care [4, 8–10]. Therefore, measuring patient experiences of care and using such data to improve care is crucial to promoting health equity.

In a program of research and knowledge mobilization known as *EQUIP Health Care*, we have been developing and testing ways to measure patient experiences of equity-oriented health care (EOHC). Grounded in a critical theoretical conceptualization of health equity, the notion of EOHC explicitly aims to: (a) address the frequent mismatches between usual approaches to care and the needs of people most impacted by health and social inequities, (b) mitigate the impacts of multiple, intersecting forms of discrimination, racism and stigma, and (c) take into account the health effects of social and structural inequities [11].

Limited attention has been given to defining what constitutes EOHC from the perspective of patients, or to developing appropriate and valid ways of measuring such care. A search of the literature shows that there are few options with regard to equity-nuanced scales to measure quality of care from the perspective of patients. Scales that aim to measure patient experiences of care, often referred to as patient-reported experience measures (PREMs), are not necessarily underpinned by an equity lens. For example, the Canadian Institute for Health Information's (CIHI) PREMs for in-patient care [12] were developed to measure experiences without explicit attention to stigma and discrimination [13], limiting the focus of PREMs to involvement in decision-making and treatment options [14]. Most measures of patient-centered care use a single variable (e.g., satisfaction on a 10-point scale, or access to care with yes/no response options). Additionally, such measures are often deployed in limited ways comparing care or outcomes of care for specific groups of people defined by a single demographic characteristic (e.g., a particular ethnocultural or racialized group) [15, 16]. Scales or items developed to measure experiences of care that strive to use a health equity lens often focus primarily on racialized groups (e.g., based on a mix of geographic origin, skin colour, and identity), and on groups experiencing financial strain [14, 17, 18]. Although measures based on variables defined by population groups can be helpful to gauge how particular groups of people experience care, within any given group there will be a diversity of intersecting factors (e.g., income, ability, gender) shaping experience. Moreover, experiences of stigma and discrimination cannot be imputed from single variables such as satisfaction or access. Experiences of care are bound up with numerous factors such as feeling that one has been treated in a respectful manner (particularly for people who have had past negative experiences), and that care has been tailored to meet their particular needs and priorities [11, 19, 20]. In this context, a broader approach is needed to measure EOHC.

While the limited evidence suggests that measuring equity in care settings is feasible through patient experience surveys [21], there are few options for measuring patient

experiences of care using an intersectional lens [10]. Intersectionality offers a perspective for understanding how multiple forms of social inequity interact and interrelate to produce relative advantages or disadvantages [22, 23]. In our research program, we have used an intersectional lens to highlight how inequities are experienced, typically, based on inter-related, co-constituted factors and conditions–and not solely based on any one particular category, variable, or group affiliation [10, 24]. For example, patients may perceive (or even anticipate) being inadequately cared for on the basis of being stigmatized as "drug using", or because they are presumed to be overusing the system (i.e., if they have sought help multiple times in a short period). As a result of the ongoing legacies of colonialism and Indigenous-specific racism in the Canadian context, Indigenous peoples are often subject to stigma related to substance use, regardless of whether or not they use substances; research shows, for example, that Indigenous peoples in Canada who come to emergency departments (EDs) and present with symptoms such as unsteady gait or slurred speech are often assumed to be using alcohol or other substances, and treated as such, when they may be experiencing a neurological condition, stroke, or other serious health issues [4, 5, 25, 26]. In many cases, people avoid seeking care or leave without being seen, due to fear of being negatively judged or treated in a dismissive manner [5, 24, 27–29]. As our team has shown in previous research, people who anticipate poor treatment report preparing for health care encounters as carefully as they can, and engaging with health care staff with vigilance and distrust, undoubtedly shaping those encounters [5, 16, 30]. The aim of efforts to measure experiences of care through an intersectional equity lens is not to determine the veracity of such reports. Rather, the aim is to illuminate how patients experience care as a constellation of inseparable and intersecting experiences that are tied to issues of power and structural conditions not amenable to being measured solely on the basis of any one variable (such as satisfaction) or demographic identifiers (such as ethnicity).

The purpose of this paper is to report on research focused on developing and testing two measures of patient experiences of care that explicitly use a health equity and intersectional lens. Our analysis focuses on two different EOHC scales. The first, the *Equity-Oriented Health Care Scale–Ongoing (EOHCS–Ongoing)* was evaluated in primary health care (PHC) settings, where patients have an ongoing relationship with providers over time. The second, the *Equity-Oriented Health Care Scale–Episodic (EOHCS–Episodic)* was evaluated in an ED, where care is provided on an episodic basis.

## Background

The program of research and knowledge mobilization known as *EQUIP Health Care* provided a foundation for developing and testing the effectiveness of EOHC interventions by first studying how care in the PHC sector was effectively provided to structurally disadvantaged and marginalized populations [31, 32]. Marginalized in this context refers to refer to the social, economic, and political conditions that contribute to health and health care inequities, and to the disproportionate effects of such inequities on particular populations [11]. Our research program focused on identifying the key dimensions of EOHC, strategies to guide organizations in implementing those key dimensions [29, 31], and identification of indicators of such care relevant in PHC settings [33, 34]. Guided by a framework articulating the key dimensions of EOHC and 10 strategies to support implementation, the *EQUIP* research team then developed an organizational-level, multi-component health equity intervention referred to as *EQUIP Primary Health Care (PHC)*, and tested it in four Canadian PHC settings [11, 19, 35]. Building on the insights from *EQUIP PHC*, we subsequently tailored and modified the intervention (referred to as *EQUIP Emergency*) to test it in three Canadian EDs [10, 36].

As evidence-based and theoretically informed interventions, *EQUIP PHC* and *EQUIP Emergency* are designed to enhance organizational capacity to provide EOHC, particularly for people who experience significant health and social inequities. Throughout *EQUIP's* work, our team has drawn on intersectionality, which emerged from Black feminist scholarship [10, 22, 37–39], and on complexity theory to draw attention to health care organizations as complex adaptive systems that are increasingly expected to be tailored and strategically redirected to meet the needs of people in varied contexts [40–42].

Partnerships and collaborations with health care agencies serving Indigenous and non-Indigenous peoples with lived and living experiences of health and social inequities are foundational to the *EQUIP Health Care* research program. Our early research with Indigenous PHC clinics provided initial data from which we developed indicators foundational to the scale development process described in the present paper [29, 31, 33–35]. For example, *EQUIP PHC* involved a partnership with an Indigenous PHC clinic that served diverse groups of Indigenous peoples, and *EQUIP Emergency* used a tripartite leadership model involving Indigenous and non-Indigenous community leaders, including an Indigenous Elder with extensive research experience as a co-PI, and leaders in practice and academic roles. Our Institutional Ethics Review Boards routinely provide an additional layer of scrutiny to ensure that research protocols comply with Canada's *Tri-Council Policy Statement 2* [43] on research involving First Nations, Inuit, and Métis peoples.

As the *EQUIP Health Care* program of research evolves, we are continually refining our understanding of the key dimensions of EOHC, which provide the basis for the *EQUIP* interventions. For the purposes of *EQUIP PHC* and *EQUIP Emergency*, these key dimensions were defined as including: (i) trauma- and violence-informed care (TVIC): recognizing and limiting the effects of trauma and violence, including structural violence, on peoples' lives and health care experiences; (ii) culturally-safe care approaches: reducing power imbalances, systemic racism, and discrimination; and (iii) harm reduction: preventing harms from substance use stigma, and in the process, promoting opportunities for well-being in the context of substance use [11]. As a cross-cutting issue, structural violence refers to the systemic disadvantages that stem from unjust structures, policies, and institutional practices, which contribute to poor health [44]. Further, in our subsequent research, our team explicitly names our stance toward cultural safety as "antiracism"[45] and integrates the concept of "Substance Use Health" as a non-stigmatizing approach that encompasses harm reduction [46, 47]. In the context of *EQUIP Health Care*, Substance Use Health is used as a lens that incorporates harm reduction, and frames substance use in relation to a spectrum inclusive of non-use, beneficial uses, occasional risks or harms, use that has ongoing or understood harms and consequences, and substance use disorders [48].

PHC settings and EDs are critical contexts within which issues of health equity and inequities must be addressed, particularly in light of ongoing reductions in community-level primary care services in most jurisdictions in Canada, with concomitant and increasing pressures on EDs to bridge the gaps in care [5, 10, 24, 49]. The literature confirms that people who experience significant health and social inequities face the greatest challenges accessing primary care; consequently, people are increasingly accessing care in EDs for needs that, if resources were available, could be addressed in the PHC sector [4–6, 26, 50–52]. For example, in previously published data from *EQUIP Emergency*, we showed that structurally disadvantaged groups of people were significantly less likely to have regular primary care access, and significantly more likely to have repeat ED visits, to present to EDs with health issues that were rated as lower acuity, and to present with chronic health problems [10]. Additionally, as discussed by other researchers, when patients who are structurally disadvantaged seek care at the ED, the chances of experiencing negative judgements or stigmatization are high [53–55]. Similarly, data from

*EQUIP Emergency* showed that structurally disadvantaged groups of people reported significantly more discrimination in EDs, and rated their care more poorly than other groups [10].

Throughout the *EQUIP* program of research, our intention has been to invite people to describe their experiences of care in ways that explicate EOHC. In the *EQUIP PHC* study, we mobilized data to develop the *EOHCS–Ongoing* (initially abbreviated to *E-HoCS*) [19]. The *EOHCS–Ongoing* was developed in PHC contexts delivering team-based care, in which there is an assumption of an ongoing relationship between patients and a health care setting. Subsequently, for the *EQUIP Emergency* study, we modified the scale to capture experiences of care in a single visit. Since care is provided on an episodic basis, this scale is called the *EOHCS–Episodic*. In this paper, we first describe the development of the original scale for use in primary care/PHC. We then discuss its ongoing adaptation for use in episodic care contexts such as EDs. We conclude with a discussion of the implications for measuring patient experiences of EOHC from an intersectional perspective, particularly in settings where patients have episodic contact across the continuum of care.

## Scale development: *Equity-Oriented Health Care Scale–Ongoing* (*EOHCS–Ongoing*) in *EQUIP Primary Health Care*

Drawing on our evolving conceptualization of the key dimensions of EOHC [11, 35], in the *EQUIP PHC* study, we used conventional scale development approaches combined with item response theory (IRT) to develop and evaluate the psychometric properties of a brief patient-reported measure of EOHC, called the *EOHCS–Ongoing*, for use in PHC and other settings where patients have ongoing contact over time. The *EOHCS–Ongoing* taps into aspects of EOHC that can be assessed using patients' self-reports. As noted above, refinements to the key dimensions of EOHC integrating findings from our ongoing research are published in Browne et al. [11] and further refinements are underway; however, the conceptual grounding of the *EOHCS–Ongoing* remains unchanged.

**Item generation, pilot testing, and item mapping.** An initial pool of 52 items was developed to reflect domains of EOHC that align with the key dimensions and would be amenable to patient self-report, drawing on our earlier research and other measurement tools in the area of patient experiences of care and quality of care [29, 31–35, 56]. Analysis of our qualitative interviews with 68 patients experiencing marginalization and accessing PHC in urban Indigenous clinics identified five domains: (1) create a welcoming, comfortable milieu (WCM); (2) promote accessibility and reduce barriers (ARB); (3) tailor care to individual context, history and experience (TIC); (4) promote emotional safety and trust (EST); and (5) convey a non-discriminatory posture (NDP). Two core/anchor items were identified for each domain. For each item, patients were asked to rate how often in the previous 12 months their PHC providers had engaged in an action reflecting EOHC, on a five-point scale ranging from "never" to "always." Cognitive interviews were conducted with five patients in one PHC setting to assess the clarity and meaning of each item from the patient's perspective and its adequacy for measuring EOHC using patient-reported experiences. Given the precarity of peoples' living situations (reliance on shelters, living on or near the street, struggling with food security, etc.), we were pleased to have obtained input from five individuals who had extensive experiences in the primary care sector. They concurred with regard to the clarity and relevance of items, and were uniformly positive about the prospect of patients being invited to respond to these items. This was not unexpected given that many of the items were adapted from existing surveys or scales and had been subjected to testing, and reflected key features of EOHC as identified by patients in our prior research. Based on this process and team analysis, 32 items, organized into five domains, were retained for psychometric testing.

In *EQUIP PHC*, we tested the *EOHCS–Ongoing* items with a cohort of patients accessing care in four PHC clinics in two Canadian provinces over two years [19]. As discussed in detail in prior publications, the cohort included a diverse set of 567 patients from four different PHC clinics mandated to serve people experiencing major structural disadvantages and marginalization (due to income, geography, education, racism, ableism, and other forms of stigma and discrimination) [11, 19, 57–59]. A detailed summary of patients' demographic characteristics is provided in a prior publication [59]; in brief, 58% identified as female, 56.3% self-identified as Indigenous, 65.4% were unemployed, 43.4% reported having less than high school education, and 36.3% reported very high levels of financial strain. The sample was recruited from patients who had an existing connection with the clinics. We invited patients to rate their overall experiences of care involving all staff, versus their impressions of any one particular staff member, realizing that primary care settings are oriented to providing team-based care.

**Psychometric evaluation of *Equity-Oriented Health Care Scale–Ongoing (EOHCS–Ongoing)*.** The structural validity of the scale was evaluated using Confirmatory Factor Analysis (CFA) in MPLUS [60] to examine the extent to which the items identified within a domain fit with the underlying construct using Chi Square and three fit indices: the Comparative Fit Index (CFI), Tucker Lewis Index (TLI) and Root Mean Squared Error of Approximation (RMSEA). Both CFI and TLI are incremental fit indices that compare a model with a baseline model (i.e. one with the worst fit); values range from 0 to 1, with a good fit indicated by values $\geq$.95 [61], with >.90 acceptable for TLI [62]. RMSEA is an absolute fit index, where a value of 0 equates to an exact fit; values of < .05 are considered *close fit;* between .05 and .08 a *fair fit;* between .08 and .10 is *mediocre fit;* and > .10 a *poor fit*. Fit and modification indices were inspected to determine whether the model fit could be improved.

With 32 items in the model arranged into five domains, the Chi-square test for the overall model was significant (see Table 1). Given our commitment to retain those domains around which the scale is organized, we then ran separate CFAs using the items in each domain to better understand how they were contributing to the latent construct, and to potentially identify items that could be deleted. For each scale, using the modification indices, and considering theory and redundancy between items, we identified eight items for deletion (at least one from each domain), resulting in 24 items remaining. Results of a new CFA conducted with these 24 items (organized into the same five domains) revealed substantially improved model fit and supported a good fit between the five-domain structure of the scale and the item pool. Within each domain, item factor loadings ranged from .47 to .91. Thus, scores for each domain and the overall scale were computed by summing applicable items and dividing by the number of items on the scale (range 0–4), where higher scores reflect more positive perceptions of EOHC.

Internal consistency was .92 for the overall scale and .65 to .82 for each domain. However, three of the five domains (ARB, EST, TIC) overlapped more than expected (correlations .92-

**Table 1. Fit indices for CFA of the *EOHCS–Ongoing*.**

| Model Tested | Fit Statistics | | | |
|---|---|---|---|---|
| | *Chi Square* | *CFI[1]* | *TLI[2]* | *RMSEA[3]* |
| Step 1: 32 items in five domains | 1964.75* | .881 | .870 | .08 |
| Step 2: 24 items in five domains | 749.85* | .934 | .932 | .06 |

Notes: CFI = Comparative Fit Index; TLI = Tucker-Lewis Index; RMSEA = Root Mean Squared Error of Approximation

* = p< .05

.96), suggesting that redundancy remained. For practical uses, the scale was also still quite long (i.e. 24 items). Further, the distribution of scores was highly skewed with limited variability. Ceiling effects are commonly observed in patient-reported experience measures focused on satisfaction with care or quality of care [63]. Ceiling effects occur because many patients rate providers at the top levels possibly because of respect or social desirability, and are problematic because they tend to disguise important differences in experiences of care, including when care is sub-optimal and in need of improvement. Thus, while these analyses supported the validity and reliability of the *EOHCS–Ongoing* based on conventional psychometric testing, we recognized that a different approach was needed to further simplify the scale and improve its ability to capture variation in perceptions of EOHC in order to be useful in research, quality improvement, or decision-making contexts.

**Improving scale precision and discrimination using IRT.** To address these issues, we drew on IRT to further reduce the length of the scale while improving its ability to discriminate between different levels of EOHC (from lowest/least ideal to highest/most ideal). As an alternative to traditional psychometric testing, IRT is ideally suited to address issues of redundancy, precision, and discrimination. IRT begins from an assumption that items measure a single domain, but are of varying levels of difficulty (rather than assuming similar difficulty of items as is the case in classical measurement theory) [64]. Here, low difficulty items would be those that providers do very frequently, while high difficulty items are those behaviors that providers are less likely to do.

Using the 24 items retained after the CFA, we used an iterative process to compare the item characteristic curves generated for each item and IRT parameters in order to select a brief pool of items that would reflect the range of difficulty across each of the five domains (analyses were conducted in STATA 16.0). In making decisions about which items to retain or delete, we privileged the two items from each scale that had been identified as core items. Results of the IRT analysis for the final 12 items in the scale are shown in Table 2, with items organized in descending order from least to most difficult. This item pool includes nine out of 10 core items and three additional items, with 10 positively-worded and two negatively-worded items. The final scale is shown in Table 2, and a formatted version has been included in S1 Table.

**Format and scoring of the 12-item *Equity-Oriented Health Care Scale–Ongoing* (*EOHCS–Ongoing*).** The *EOHCS–Ongoing* is comprised of 12 items that reflect five domains of EOHC. Respondents are asked to rate, in the previous 12 months, the extent to which their interactions with health care staff were equity-oriented on a five-point scale including never (0), rarely (1), sometimes (2), usually (3), and always (4). The *EOHCS–Ongoing* total score is a count of the number of items rated by patients as "always" occurring (for 10 positively worded items) and "never" occurring (for two negatively worded items), with a range of 0 to 12. Scores on the *EOHCS–Ongoing* provide an index of the degree or level of EOHC, from lower to higher. Total scores correlated with single-item measures of overall quality of care (r = .602) and fit of care with needs (r = .599), providing evidence of concurrent validity. Overall quality of care was adapted from a global satisfaction with health care item developed by Nápoles et al. [65]. Fit of care, developed for *EQUIP PHC* and informed by existing literature, was phrased as, "In the past 6 months, how well has the help you have received at the clinic fit with your needs?" with four response options (not well, somewhat well, well, very well). In health services research, the fit between care provided, and the alignment of such care with patients' needs in the context of their lives, are considered important features of quality of care [66]. These served as standard measures against which the *EOHCS–Ongoing* was validated.

In the *EQUIP PHC* study, as previously published [19], patients completed structured interviews, which included the *EOHCS–Ongoing* and other self-reported health outcome and quality of life scales, at four time points (baseline, 12, 18, and 24 months later). As discussed in the

**Table 2. Item difficulty and rates of item endorsement by participants in IRT analysis.**

| Item | Difficulty | % Endorsed |
|---|---|---|
| **In the past 12 months, how often have you felt discriminated against by staff here, including health care providers, receptionists and others?** | -1.71 | 88% |
| **In the past 12 months, how often did your health care providers have a negative attitude toward people using services because of mental health concerns?** | -1.61 | 84% |
| **In the past 12 months, how often did staff here treat you with courtesy and respect?** | -1.12 | 79% |
| **In the past 12 months, how often did your health care providers try to make you feel as comfortable as possible?** | -1.03 | 81% |
| **In the past 12 months, how often did the staff here welcome you when you came for care?** | -0.95 | 73% |
| **In the past 12 months, how often did your health care providers encourage you to come and see them or call when you need to?** | -0.71 | 71% |
| **In the past 12 months, how often did your health care providers seem open to talking about sensitive issues, for example, grief, mental health problems, substance use, or abuse experiences?** | -0.45 | 63% |
| **In the past 12 months, how often did your health care providers help you to work on any barriers you have accessing health care (e.g., costs of medication or services, problems with transportation or childcare, problems getting a referral, etc.)?** | -0.14 | 54% |
| **In the past 12 months, how often did your health care providers give you health advice that is suitable for your everyday life?** | -0.03 | 51% |
| **In the past 12 months, how often did your health care providers try to help you to get services that are not offered here (such as social assistance, disability benefits, housing, or parenting support)?** | 0.21 | 44% |
| **In the past 12 months, how often did your health care providers ask you about who is important in your life?** | 0.69 | 28% |
| **In the past 12 months, how often did your health care providers ask about basic resources that affect your health, such as food, clothing, or shelter?** | 0.76 | 27% |

*Instructions: These questions ask about your experiences with staff at this service site in the past 12 months. By staff, we mean anyone who works here including health care providers, reception staff, and others.

prior publication, using path analysis techniques, longitudinal analysis with the *EOHCS–Ongoing* showed that providing more EOHC predicted improvements in important patient health outcomes 18 months later, supporting predictive validity of the *EOHCS–Ongoing*. Collectively, psychometric evaluation suggests that the *EOHCS–Ongoing* is a promising measure of EOHC that may be useful in assessing the possible impacts of interventions to enhance EOHC in PHC settings or in monitoring care delivery as part of Continuous Quality Improvement (CQI).

## Scale adaption and development of the *Equity-Oriented Health Care Scale–Episodic (EOHCS—Episodic)* in *EQUIP Emergency*

The construct of EOHC underpinning *EQUIP PHC*, and *EQUIP Emergency*, and the theoretical approach underlying these studies were consistent [24, 36]. Thus, we considered that many of the items in the *EOHCS–Ongoing* would be relevant to measuring patient's reported experiences of care. However, given the different relationships that patients have in relation to accessing care in two different health care contexts–PHC and EDs–the response options and time frame needed to be adapted to reflect their experiences during a single, episodic visit.

**Item generation, testing, and mapping.** Using the *EOHCS–Ongoing* developed for the PHC context, our research team members, who had worked with the theoretical underpinnings of EOHC for decades (including those with expertise in emergency care) adapted the items to suit the episodic care setting. Each *EOHCS–Ongoing* item was reviewed through the

**Table 3. Original experiences of *EOHCS–Episodic* scale items.**

| During this visit, did staff: | Yes | No |
|---|---|---|
| 1. make you feel welcome? | ○ | ○ |
| 2. try to make you as comfortable as possible? | ○ | ○ |
| 3. treat you with courtesy and respect? | ○ | ○ |
| 4. discriminate against you? | ○ | ○ |
| 5. seem open to talking about what is important to you? | ○ | ○ |
| 6. learn enough about you to give useful advice? | ○ | ○ |
| 7. give you advice that is suitable for you? | ○ | ○ |
| 8. learn about problems you might have getting services (e.g., costs, transportation, getting a referral, etc.)? | ○ | ○ |
| 9. try to help you get services you need? | ○ | ○ |
| 10. encourage you to return if you need to? | ○ | ○ |

lens of emergency and episodic care. The research team also worked with clinical practice leads at two ED sites to confirm whether the items would work in an episodic context. Two items eliminated from the *EOHCS–Ongoing* in the development of the *EOHCS–Episodic* were "ask about basic resources that affect your health" and "have a negative attitude toward people using services because of mental health concerns". The former was judged to be beyond the scope of usual episodic and ED practice, and the latter was too specific, with discrimination in general being a broader issue, and captured by the fourth item. A comparison of the *EOHCS–Ongoing* and *EOHCS–Episodic* items shows how, for the episodic context, we broadened from primary care specific issues. We also simplified and streamlined the questions. A review by diverse stakeholders suggested that using a five-point scale would require a level of discernment not easily made during an episodic, and often brief, health care encounter; consequently, we changed the response option to a simple yes/no. Ten adapted items were used with the binary response option for our initial testing in *EQUIP Emergency*, as shown in Table 3.

In the context of *EQUIP Emergency*, and embedded in the larger patient survey, we tested these 10 items at one of our three hospital sites, Surrey Memorial Hospital (SMH), during a wave of patient surveys [10, 24, 36]. Recruitment for the larger *EQUIP Emergency* study began on November 28, 2017, and ended on November 12, 2020. The study protocol was approved by the research ethics boards of the University of British Columbia, the University of Northern British Columbia, and the research ethics boards of Fraser Health Authority, Northern Health Authority, and Providence Health Care (approval #s H16-03397, H17-01548, and H18-01423). Consent was documented on signed consent forms for all participants.

SMH is the largest ED in the Canadian province of British Columbia, and serves diverse suburban communities, including high proportions of newcomers, with many who speak a language other than Canada's two official languages, English or French, at home [67, 68]. In Canada, "newcomers" is the preferred term to indicate people who were not born in Canada; this includes people classified by the Canadian federal government as immigrants or refugees [69]. The hospital also serves the largest urban Indigenous population among municipalities in BC, with 2.16% (16,300 people) of the population self-identifying as First Nations, Métis, or Inuit [70–73]. Due to COVID-19 pandemic restrictions and mandated requirements to halt data collection, it was not possible to administer the *EOHCS–Episodic* at the two other ED sites involved in the *EQUIP Emergency* study.

Surveys were conducted with 284 patients, during which they were asked about their experiences of receiving care during their visit to the ED. Research Assistants, trained in equity-oriented approaches including strategies for working respectfully with people who experience

significant inequities (and are thus often not included in research), conducted the surveys and gathered patients' feedback on the clarity, meaning, and response options to the items. The recruitment efforts resulted in a sample that was diverse and was generally representative of the populations served by SMH. This included representation from people over 65, Indigenous peoples, people experiencing precarious housing, people born outside of Canada, and people who found it difficult to live on their income (Table 4). Compared to provincial population trends, the sample tended to be more diverse in terms of having greater representation from those born outside of Canada, people who did not have English as their first language, as well as greater representation from people who self-identified as Indigenous. The entire sample is described more fully elsewhere [10].

The sample collected at SMH included 48.4% of respondents who reported they were not born in Canada, and 42.3% who reported a first language other than English. In addition, 50.2% of the respondents reported at least somewhat or very difficult financial strain as measured by the Financial Strain Index (Table 4) [75]. Consistent with research that indicates that patients in Canadian health care settings tend to rate ED care favorably, overall [10, 50, 76], ratings of quality of care in our sample were high, with an average of 8.11 (SD 2.14) on a scale of 0–10, as measured by an item from the British Columbia Emergency Department Patient Experiences of Care scale [10, 24, 36, 77]. As reported elsewhere, in an effort to understand experiences of care in a more nuanced way, and in light of deepening health care inequities in Canada [26, 78, 79], the larger *EQUIP Emergency* survey also sought to understand patients' experiences of discrimination, both in their everyday life and during their ED visit [10, 24, 36]. In the study sample, using the Everyday Discrimination Scale [80], 63% reported experiencing some form of discrimination in their everyday lives, however, on a scale of 0–45, the mean was relatively low at 8.36 (SD 9.421). Similar to trends seen in the larger study, structurally disadvantaged groups of people reported significantly more discrimination in the SMH ED.

**Psychometrics: *Equity-Oriented Health Care Scale–Episodic (EOHCS–Episodic).*** The *EOHCS–Episodic* items reflected the key dimensions of EOHC as discussed above. Patients were asked during their visit whether their interactions with health care staff were equity-oriented, using a binary response scale: yes (1) or no (0). The *EOHCS–Episodic* total score is a count of the number of items rated by patients as "yes" (1) for all items except "discriminated against you" which received 1 point in the count for "no" responses, with a range of 0 to 9. Scores on the *EOHCS–Episodic* provide an index of the degree or level of episodic EOHC, from lower to higher.

IRT with a two-parameter (difficulty and discrimination) model was used to examine the item characteristics of the 10 items in the *EOHCS–Episodic*. To test the concurrent validity of the *EOHCS–Episodic*, correlations of the total *EOHCS–Episodic* score and quality of care and t-tests of differences in *EOHCS–Episodic* total scores by individual characteristics (age, gender, financial situation, identity as Indigenous or non-Indigenous, and employment status) were conducted. All analyses were conducted in STATA 15.0.

In the IRT analysis of the 10 items of the *EOHCS–Episodic*, one item ("Encourage you to return, if you need to") had a discrimination score of 0.99. Discrimination scores close to 1.0 indicate that the item does not differentiate between people with varying degrees of the underlying concept. This item's poor discrimination is likely related to the fact that the item does not apply to all people in the context of care provided in EDs, and so was dropped and the two-parameter model was re-estimated. Table 5 shows the nine-item *EOHCS–Episodic*, and provides the final model with difficulty and percentage of people endorsing each item. As difficulty scores for items decreased, more participants responded "yes" with respect to that item. The final version of the scale can be found in S2 Table.

**Table 4. Demographic characteristics of patients completing *EOHCS–Episodic* (N = 284).**

| Variable | n (%) of *EQUIP Emergency* Sample | n (%) of BC Census Sample [74] |
|---|---|---|
| **Age** | Range: 18–96, Mean: 48.97, SD: 18.529 | Range: 0–100+, Mean: 42.3, Median: 43.0 |
| **Age 65 and over** | | |
| Under 65 | 217 (77.2) | 3799070 (81.7) |
| Over 65 | 64 (22.8) | 848985 (18.3) |
| **Gender** | | |
| Woman | 147 (51.9) | 2369815 (51.0) |
| Man | 135 (47.7) | 2278245 (49.0) |
| Non-binary | 1 (0.4) | NA |
| **Education** | | |
| Didn't complete secondary school / high school | 44 (15.7) | 601640 (15.5) |
| Completed secondary school / high school | 54 (19.3) | 1138565 (29.4) |
| Some or completed post-secondary | 182 (65.0) | 2130175 (55.0) |
| **Born in Canada** | | |
| No | 137 (48.4) | 1292675 (30.5) |
| Yes | 146 (51.6) | 3167155 (69.5) |
| **First language English** | | |
| No | 116 (42.3) | 1428305 (31.1) |
| Yes | 158 (57.7) | 3170110 (68.9) |
| **Speaks English** | | |
| Does not currently speak English | 15 (5.3) | 151760 (3.4) |
| Currently speaks English | 269 (94.7) | 4442695 (96.6) |
| **Indigenous** | | |
| Non-Indigenous | 259 (91.5) | 4289655 (94.1) |
| Indigenous | 24 (8.5) | 270585 (5.9) |
| **Living situation** | | NA |
| Precarious housing | 12 (4.2) | |
| Stable housing | 272 (95.8) | |
| **Accessed a shelter in the past year** | | NA |
| No | 278 (97.9) | |
| Yes | 6 (2.1) | |
| **Primary work status** | | |
| Employed FT or PT | 138 (48.8) | 2305690 (59.6) |
| Unemployed | 62 (21.9) | 165975 (4.3) |
| Retired | 67 (23.7) | 1398710 (36.1) |
| Other (includes seasonal, exchange services or student) | 16 (5.7) | NA |
| **Receiving social assistance** | | NA |
| Not receiving | 268 (94.4) | |
| Receiving | 16 (5.6) | |
| **Receiving disability benefits** | | NA |
| Not receiving | 235 (82.7) | |
| Receiving | 49 (17.3) | |
| **Difficulty living on income** | | NA |
| Very difficult | 47 (16.7) | |
| Somewhat difficult | 94 (33.5) | |

*(Continued)*

**Table 4.** (Continued)

| Variable | n (%) of *EQUIP Emergency* Sample | n (%) of BC Census Sample [74] |
|---|---|---|
| Not very difficult | 62 (22.1) | |
| Not at all difficult | 78 (27.8) | |
| **Experience any discrimination in everyday life** | | NA |
| No | 105 (37.0) | |
| Yes | 179 (63.0) | |
| **Overall health** | | NA |
| Poor | 47 (16.6) | |
| Fair | 64 (22.6) | |
| Good | 96 (33.9) | |
| Very good | 46 (16.3) | |
| Excellent | 26 (9.2) | |
| **ED visits in the last 6 months** | Range: 1–180, Mean: 2.93, SD: 11.273 | NA |
| One visit | 793 (48.7) | |
| More than one visit | 834 (51.3) | |
| **Have usual primary care home** | | NA |
| No | 159 (58.7) | |
| Yes | 112 (41.3) | |

Total scores on the *EOHCS–Episodic* ranged from 0 to 9, with a median of 7.5. Cronbach's alpha for the nine items was 0.82. Evidence of concurrent validity, based on a high correlation with quality of care was strong (r = 0.61). Table 6 summarizes the differences in *EOHCS–Episodic* scores by participant characteristics.

As shown in Table 6, greater financial strain was associated with lower *EOHCS–Episodic* scores. Those with lower *EOHCS–Episodic* scores were more likely to self-identify as Indigenous, to have a recent shelter stay, to have English as their first language, and to be unemployed. Scores did not vary by age or gender. The lower scores among people who have English as their first language may be explained, in part, by research showing that newcomers whose first language is not English tend to rate their satisfaction with care quite highly, reflecting some peoples' appreciation for access to care that may be less accessible in their countries of origin [81–84]. In the course of data collection in *EQUIP Emergency*, respondents

**Table 5. Item difficulty parameters from the IRT model and the frequency of each item endorsed.**

| | Difficulty | % Responding Yes |
|---|---|---|
| **1. discriminate against you?** | -1.79 | 8.8% |
| **2. treat you with courtesy and respect?** | -1.44 | 93.3% |
| **3. make you feel welcome?** | -1.43 | 89.9% |
| **4. try to make you as comfortable as possible?** | -1.09 | 81.9% |
| **5. give you advice that is suitable for you?** | -0.96 | 80.7% |
| **6. seem open to talking about what is important to you?** | -0.81 | 77.3% |
| **7. learn enough about you to give useful advice?** | -0.81 | 76.1% |
| **8. try to help you get services you need?** | -0.69 | 68.1% |
| **9. learn about problems you might have getting services (e.g., costs, transportation, getting a referral, etc.)?** | 0.45 | 39.1% |

commented comparing the challenges they experienced accessing care in their countries of origin. It is also possible that, given the high proportion of newcomers and people with a first language other than English (49.4%) residing in this hospital's catchment area, the SMH ED was already making efforts to be responsive to newcomers whose first language is not English [85]. Research shows that once newcomers are able to access care, they tend to rate their care received in Canada highly and report high levels of trust in their providers [81–84]. Thus, this pattern of higher *EOHCS–Episodic* scores among people with a first language other than English aligns with published literature illustrating overall high ratings of care among newcomer groups, many for whom English is not their first language [68, 85]. With regard to the lower *EOHCS–Episodic* scores experienced by Indigenous peoples, these findings are not surprising in the Canadian context and align with ongoing research demonstrating the extent to which high proportions of Indigenous peoples face multiple forms of discrimination and stigma when accessing health care, impacting their experience of care [4, 5, 8, 68, 86].

## Discussion & implications for both scales

We have systematically used an equity lens to study people's experiences of care, and from that identified the key dimensions of EOHC, and derived items to measure experiences that align with those key dimensions [11, 19]. Through our research, we have developed two scales that serve as PREMs: one for use in settings where the care relationship is ongoing or longitudinal,

**Table 6. Differences in *EOHCS–Episodic* total score by participant characteristics.**

| | Mean (SD) | p-value |
|---|---|---|
| **Living situation** | | |
| Precarious | 6.00 (2.93) | .141 |
| Stable | 7.02 (2.21) | |
| **Financial Strain** | | |
| At least somewhat difficult | 6.66 (2.44) | .028 |
| Not very/not at all difficult | 7.28 (1.99) | |
| **Age** | | |
| Under 65 | 6.92 (2.28) | .614 |
| 65 or older | 7.10 (2.16) | |
| **Gender** | | |
| Female | 7.00 (2.11) | .911 |
| Male | 6.97 (2.38) | |
| **First Language** | | |
| Not English | 7.29 (1.94) | .044 |
| English | 6.70 (2.45) | |
| **Indigenous** | | |
| No | 7.14 (2.06) | < .001 |
| Yes | 5.10 (3.22) | |
| **Shelter in Last 6 months** | | |
| No | 7.02 (2.20) | .029 |
| Yes | 5.00 (3.35) | |
| **Employment** | | |
| Employed | 7.05 (2.16) | .034 |
| Unemployed | 6.37 (2.72) | |
| Retired | 7.45 (1.86) | |

such as in primary care, and another for use in settings where care is episodic, such as in EDs, walk-in clinics, urgent care centers, outpatient departments, among others.

There continues to be strong calls to integrate attention to equity in health care provision, and emerging evidence of positive impacts for patients, providers, and organizations. However, without access to brief and reliable ways of measuring whether equity-enhancing innovations have the intended impacts, and for whom, organizations will continue to face significant challenges justifying and funding such initiatives. Both the *EOHCS–Ongoing* and *EOHCS–Episodic* are brief, easy-to-administer patient self-report scales that can support organizations to effectively monitor quality of care using an intersectional equity lens. Embedding such scales in CQI processes and tracking responses over time, may support, shift or expand the ways in which quality of care is conceptualized, defined, enhanced, and measured in health care systems. For example, in a recent CQI initiative at a primary care clinic serving women experiencing significant social disadvantages and marginalization, items from the *EOHCS–Ongoing* were used to assess women's perspectives regarding the quality of care received [87]. The analysis of women's responses was particularly useful in highlighting those aspects of care that women rated most highly (e.g., promoting emotional safety and trust), thus helping clinicians and team-based care providers to identify domains of EOHC they seemed to excel in, as well as areas of care needing further improvement (e.g., assisting patients with housing and food security issues that impact health). While the *EOHCS–Ongoing* was developed for use in PHC care settings delivering team-based care, it may be appropriate in settings where care is provided by individual providers, for example, physicians or nurse practitioners, where the goal is to provide care over time with a roster of patients.

Both scales are potentially useful in identifying areas of care redesign to prioritize for improvements, and for monitoring the impact of actions to improve care, particularly for patients most impacted by health and social inequities. For example, in *EQUIP PHC*, the *EOHCS–Ongoing* was valuable in identifying how highly patients rated their experiences of feeling accepted and not judged by clinic providers and staff. Items rated less favorably by patients, such as how often providers helped to address barriers to accessing health care (e.g., costs of medications, problems with transportation or childcare, problems getting referrals) can also be used to generate discussion, strategies, and actions to strengthen these aspects of EOHC. Drawing attention to these specific aspects of EOHC may be particularly powerful, given the extent to which they are often considered to be outside the purview of health care. Importantly, as health care teams and organizations endeavour to improve patient experiences of care, efforts must be taken to avoid equity-oriented data collection efforts that are primarily performative (i.e., fulfilling "check-box" approaches to performance-monitoring) without action-oriented commitments, including buy-in from leadership to support actual improvements in EOHC and delivery approaches.

Given that the items in both scales focus on experiences of care in varied settings, we have also explored the use of these measures beyond health care services. For example, in a study assessing the integration of TVIC as a key dimension of EOHC in educational contexts, we added relevant items from the *EOHCS–Ongoing* to the Attitudes Related to Trauma Informed Care Scale (ARTIC), a pre-existing scale assessing trauma-informed practice [88, 89]. Indeed, as found in a recent scoping review [90], a key limitation of measures to assess EOHC concepts, such as TVIC, is that they rarely include items focused on stigma, racism, and discrimination as structural or systemic influences on care experiences. We therefore encourage the testing and use of the *EOHCS–Ongoing* and the *EOHCS–Episodic* as measures of EOHC beyond health care settings.

In view of ongoing constraints in Canadian health care sectors, diminishing opportunities for continuity of care, and increasing shifts toward episodic health care delivery, the

availability of both scales makes it possible to measure EOHC in a wide range of contexts. These ways of describing, assessing, and measuring patient experiences of care will be important for research, quality improvement and monitoring, and potentially for informing ongoing health care reforms. Both scales have the capacity to shed light on experiences of care using an intersectional lens–providing a more nuanced understanding of EOHC–in contrast to focusing on a single dimension. The use of these scales can highlight how peoples' intersecting social locations impact their experiences of care. For example, the *EOHCS–Episodic* allowed us to examine those aspects of care that people valued most highly in EDs [10]. Bringing a health equity lens to analyses of patient experiences of care is especially important to inform strategies to enhance care and reduce persistent health equity gaps. Further testing will enable ongoing refinements to both scales, and provide important insights regarding their acceptability, feasibility, reliability, and validation in diverse settings.

## Supporting information

**S1 Table. Equity-Oriented Health Care Scale–Ongoing.** A final formatted version of the *EOHCS–Ongoing*, including scoring information.
(DOCX)

**S2 Table. Equity-Oriented Health Care Scale–Episodic.** A final formatted version of the *EOHCS–Episodic*, including scoring information.
(DOCX)

## Acknowledgments

We extend our gratitude to the many patients, staff and organizations with whom we have collaborated on research in the PHC and ED sectors. Thank you to Cheyanne Stones for working so ably with our research teams.

## Author Contributions

**Conceptualization:** Annette J. Browne, Colleen Varcoe, Marilyn Ford-Gilboe, C. Nadine Wathen.

**Formal analysis:** Nancy Perrin.

**Funding acquisition:** Colleen Varcoe.

**Investigation:** Annette J. Browne, Colleen Varcoe, Marilyn Ford-Gilboe, C. Nadine Wathen.

**Methodology:** Nancy Perrin.

**Writing – original draft:** Annette J. Browne, Colleen Varcoe, Nancy Perrin.

**Writing – review & editing:** Annette J. Browne, Colleen Varcoe, Marilyn Ford-Gilboe, C. Nadine Wathen, Erin Wilson, Vicky Bungay, Nancy Perrin.

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
