## [Decision Letter · Decision Letter 0]

25 Mar 2024

PONE-D-23-43135Using a health equity lens to measure patient experiences of care in diverse healthcare settingsPLOS ONE

Dear Dr. Browne,

Thank you for submitting your manuscript to PLOS ONE. After careful consideration, we feel that it has merit but does not fully meet PLOS ONE’s publication criteria as it currently stands. Therefore, we invite you to submit a revised version of the manuscript that addresses the points raised during the review process.

We look forward to receiving your revised manuscript.

Kind regards,

Masoud Behzadifar

Academic Editor

PLOS ONE

Journal Requirements:

2. In the online submission form, you indicated that The datasets generated and/or analyzed during the current study are not publicly available because we are still actively working on analyses, but are available from the corresponding author on reasonable request.

Reviewers' comments:

Reviewer's Responses to Questions

**Comments to the Author**

1. Is the manuscript technically sound, and do the data support the conclusions?

Reviewer #1: Yes

Reviewer #2: Yes

2. Has the statistical analysis been performed appropriately and rigorously? 

Reviewer #1: N/A

Reviewer #2: Yes

3. Have the authors made all data underlying the findings in their manuscript fully available?

Reviewer #1: Yes

Reviewer #2: No

4. Is the manuscript presented in an intelligible fashion and written in standard English?

Reviewer #1: Yes

Reviewer #2: Yes

5. Review Comments to the Author

**Reviewer #1:** Summary

The is a well-presented paper that documents methods and outputs from work done to address a gap in the lack of tools for assessing patient reported experiences of care which are equity centric. Authors report on their process for developing and piloting two different patient experiences of care measures (one for primary care and the other for emergency care) that explicitly use a health equity and intersectional lens and that can be adopted by organizations to monitor quality of care. Importantly, authors have taken a perspective of equity that is informed by critical theory and rightly position equity as a multi-dimensional outcome that requires differential strategies. The primary health care scale was tested with a cohort of 567 patients while the second scale was tested in an emergency department setting with 284 patients. I appreciate the timeliness and relevance of this work and how it builds upon long standing conversations in this space.

Comments

1. The Abstract reads: “The primary health care scale, tested with a cohort of 567 patients, showed that providing more equity-oriented health care predicted improvements in important patient self-report health outcomes over time.” My understanding is that this is related to previous work (EQUIP PHC) where repeat surveys were conducted with the EHoCS. Would recommend focusing the abstract on the net new work that this paper is positioned to describe/report, except if being reported for the background section.

2. “….aim is to illuminate how indicators of PEOC tend to be experienced as a constellation of inseparable and intersecting experiences that are tied to issues of power and structural conditions not amenable to being measured solely on the basis of any one variable or characteristic (such as ethnicity).” Can authors provide some guidance in the discussion section on how users of either of the developed tools can move beyond equity-oriented data collection as a performance check box, to action-oriented utility? Otherwise, the tools may be (in)advertently misappropriated to maintain the status quo while giving originations a pat on the back for data collection.

3. In the section “Format and scoring of the 12-items EHoCS”, a sentence reads: “Total scores correlated with measures of overall quality of care (r= .602) and fit of care with needs (r = .599), providing evidence of concurrent validity.” What standard measures of quality of care and fit of care with needs was the scale being validated against? Stating this would help buttress the point on concurrent validity of the scale.

4. It was not clear why on the one hand, authors acknowledge that research consistently shows that ED care in Canada is overall rated favourably (citing references, 8, 40 and 59)- page 17, but on page 19, the rationale for why newcomers whose first language is not English, rate ED visits favourably, is out of comparative appreciation related to deficits in their home countries. Seeing as the paper acknowledges that the study site- SMH serves a higher proportion of newcomers, it is perhaps more likely that they are already more advanced in providing equitable and responsive care to this population group- one that is anchored within a trauma-based lens and considers the social determinants of health.

Others

1. I am curious to know if this project involved working with Indigenous communities/patients, and if so, where there are specific protocols and considerations applied?

2. Authors have chosen to use the term PEOC (patient experiences of care), which is not commonly used. Why not use the more commonly known term- PREM (Patient reported experience measure) which will not only promote engagement with this important piece of work, but also enhance its visibility?

**Reviewer #2:** This article presents two new measures of patient experiences of care through an equity lens, one for primary care settings, and one for episodic care obtained in emergency departments. While patients’ experiences in healthcare settings are relevant to understanding and addressing persistent population health inequities, current standardized measures of patient experience do not adequately address issues of equity. Therefore, this presentation of two new scale measures is important to the field in terms of both practice improvement and continuing research. Overall, this article is well-written and the statistical approaches appear to be adequate. I recommend publication pending relatively minor revisions. Specific comments follow:

1. While the authors present two new scales, one to some degree an adaptation of but also quite different from the other, the presentation of scale development is inconsistent between the two. For example, survey participant demographics are included in the narrative about developing the episodic EEE-HC scale, but not for the primary care EHoCS scale. Please provide all relevant demographic and analytic summary statistics for both the 5-point and 2-point scales and clearly explain any differences in approach.

2. Page 7, line 150 and line 154, and throughout. Equity-Oriented Health Care Scale actually translates to an acronym of EOHCS or EoHCS, not EHoCS. I see how EHoCS is easier to pronounce as a word, but the mis-ordered letters are confusing. The episodic care scale, EEE-HC, is quite different, although it attempts to measure similar constructs in a different setting; it is also not easily pronounced. You might consider using the same stem for each scale, with a different setting indicator, e.g., xxxx-PCC for primary care clinic and xxxx-ECED for episodic care in emergency departments. Ideally, the scale name should clearly indicate it is a measure of patient experience of equitable care.

3. Page 6, line 120. If the term “structural violence” is to be used, provide a citation and explanation of meaning.

4. Page 8, line 178. Five cognitive interviews seems to be a fairly low number for scale development. Why was this number deemed to be adequate? Was saturation achieved this quickly?

5. Page 11, Table 2. The question stem does not align with some of the items, specifically those that repeat “how often…” from the question stem.

6. Page 12, line 251. Please provide wording for all values of the response options, including those between never (0) and always (4).

7. Page 12, lines 259-262. As worded here, the narrative seems to imply that path analysis produces causal inferences, which it does not. Such analysis may support predictive validity statistically and is certainly useful to general hypotheses of causality to be further researched. However, it does not indicate that the scale is a “measure” of benefits.

8. Provide the final version of each scale, including all question stems, items, response options, and scoring.

6. PLOS authors have the option to publish the peer review history of their article (what does this mean?). If published, this will include your full peer review and any attached files.

Reviewer #1: No

Reviewer #2: No

---

## [Author Response · Author response to Decision Letter 0]

24 Apr 2024

Please find the authors’ responses in blue font below, and corresponding line numbers on the revised manuscript with tracks in orange the (a) comments re journal requirements, and (b) the two sets of reviewers’ comments. 

Journal Requirements: 

In the online submission form, you indicated that “The datasets generated and/or analyzed during the current study are not publicly available because we are still actively working on analyses, but are available from the corresponding author on reasonable request.”

All PLOS journals now require all data underlying the findings described in their manuscript to be freely available to other researchers, either a. In a public repository, b. Within the manuscript itself, or c. Uploaded as supplementary information. This policy applies to all data except where public deposition would breach compliance with the protocol approved by your research ethics board. If your data cannot be made publicly available for ethical or legal reasons (e.g., public availability would compromise patient privacy), please explain your reasons on resubmission and your exemption request will be escalated for approval. 

The datasets generated and/or analyzed for this study are not publicly available due to issues of safety, sensitivity and privacy. Participants in this study consented only to sharing of their research data within the immediate team and not with external parties. Public deposition would breach compliance with the protocols approved by our research ethics boards. Inquiries about access to data should be directed to the Principal Investigators (Annette J. Browne and Colleen Varcoe). 

Please include your full ethics statement in the ‘Methods’ section of your manuscript file. In your statement, please include the full name of the IRB or ethics committee who approved or waived your study, as well as whether or not you obtained informed written or verbal consent. If consent was waived for your study, please include this information in your statement as well. 

A full ethics statement has been added in lines 359-363. We have included the full names of the IRBs who approved our study and the approval-certificate numbers.

Reference List: we have reviewed our reference list to ensure it is complete and correct. We have also added additional references in text and to the reference list. 

Reviewer 1’s Comments and Authors Responses:

1. The Abstract reads: “The primary health care scale, tested with a cohort of 567 patients, showed that providing more equity-oriented health care predicted improvements in important patient self-report health outcomes over time.” My understanding is that this is related to previous work (EQUIP PHC) where repeat surveys were conducted with the EHoCS. Would recommend focusing the abstract on the net new work that this paper is positioned to describe/report, except if being reported for the background section.

Thank you for this comment. We have revised the sentence above to say, “The primary health care scale was first tested with a cohort of 567 patients.” We think that this revision will provide readers with sufficient background regarding our prior, which informs the new work presented in this paper. Please see lines 33-34.

2. “….aim is to illuminate how indicators of PEOC tend to be experienced as a constellation of inseparable and intersecting experiences that are tied to issues of power and structural conditions not amenable to being measured solely on the basis of any one variable or characteristic (such as ethnicity).” Can authors provide some guidance in the discussion section on how users of either of the developed tools can move beyond equity-oriented data collection as a performance check box, to action-oriented utility? Otherwise, the tools may be (in)advertently misappropriated to maintain the status quo while giving originations a pat on the back for data collection.

Thank you for this important comment and suggestion. In the Discussion, in lines 485-498 we have provided examples of how the scales can be useful in identifying and monitoring areas for improvement. We have also cautioned against using the scales as part of equity-oriented data collection efforts that are primarily performative in intent (i.e., fulfilling ‘check-box’ approaches to performance-monitoring) without action-oriented commitments including leadership buy-in to support actual improvements in equity-oriented care and delivery approaches.

3. In the section “Format and scoring of the 12-items EHoCS”, a sentence reads: “Total scores correlated with measures of overall quality of care (r= .602) and fit of care with needs (r = .599), providing evidence of concurrent validity.” What standard measures of quality of care and fit of care with needs was the scale being validated against? Stating this would help buttress the point on concurrent validity of the scale.

We have clarified that quality of care was measured using a single item adapted from the global satisfaction with health care measure developed by Nápoles et al. (2009), and that ‘fit of care with needs’ was developed based on existing literature for the EQUIP PHC study. These served as standard measures against which the EOHCS was validated. These points have been added to the paragraph in lines 313-318. 

4. It was not clear why on the one hand, authors acknowledge that research consistently shows that ED care in Canada is overall rated favourably (citing references, 8, 40 and 59)- page 17, but on page 19, the rationale for why newcomers whose first language is not English, rate ED visits favourably, is out of comparative appreciation related to deficits in their home countries. Seeing as the paper acknowledges that the study site- SMH serves a higher proportion of newcomers, it is perhaps more likely that they are already more advanced in providing equitable and responsive care to this population group- one that is anchored within a trauma-based lens and considers the social determinants of health.

In lines 444-453 we have provided additional explanations to contextualize the scores among newcomers. We have clarified that, according to our review of the literature, newcomers whose first language is not English tend to rate their satisfaction with care quite highly, reflecting some peoples’ appreciation for access to care that may be less accessible in their countries of origin. We have also noted that, in the course of data collection in EQUIP Emergency, newcomers whose first language was not English commented on the comparative challenges they experienced accessing care in their countries of origin. We have also added that it is possible that, given the high proportion of newcomers residing in this hospital’s catchment area, that the emergency department was already making efforts to be responsive to newcomers who first language is not English. 

5. I am curious to know if this project involved working with Indigenous communities/patients, and if so, where there are specific protocols and considerations applied? 

Yes, partnerships and collaborations with healthcare agencies serving Indigenous and non-Indigenous people with lived experiences of health and social inequities are foundational to EQUIP Health Care. We have explained that EQUIP PHC involved a partnership with an Indigenous primary health care clinic whose mandate is to provide care to diverse groups of Indigenous peoples, and EQUIP Emergency used a tripartite leadership model involving Indigenous/community, practice, and research team members, including an Indigenous Elder with extensive research experience. Our IRBs routinely provide an additional layer of scrutiny to ensure that research protocols comply with Canada’s Tri-Council Policy Statement 2 (2022) on research involving First Nations, Inuit, and Métis peoples. Thank you for the suggestion that we explain this explicitly. We have done so in the Background in lines 146-155. 

6. Authors have chosen to use the term PEOC (patient experiences of care), which is not commonly used. Why not use the more commonly known term- PREM (Patient reported experience measure) which will not only promote engagement with this important piece of work, but also enhance its visibility?

Thank you for this valuable input. We now spell in full the phrase ‘patient experiences of care’ in the context of the sentences where the acronym ‘PEOC’ was used. It is now unnecessary to use PEOC as an acronym in any instance. We also make reference to PREMs as one way to conceptualize such measures of patient experiences of care. We also note in the paper that the scales we present are examples of PREMs (in line 73 and in line 462). We have also added PREM to list of key words. 

Reviewer 2’s Comments and Authors’ Responses:

1. While the authors present two new scales, one to some degree an adaptation of but also quite different from the other, the presentation of scale development is inconsistent between the two. For example, survey participant demographics are included in the narrative about developing the episodic EEE-HC scale, but not for the primary care EHoCS scale. Please provide all relevant demographic and analytic summary statistics for both the 5-point and 2-point scales and clearly explain any differences in approach.

Thank you for this recommendation. We have provided all relevant demographic and summary statistics for the samples used in both scales, including reference to where a detailed description of the sample (n=567) for the 5-point scale was previously published in BMJ-Open. Please see lines 236-239.

2. Page 7, line 150 and line 154, and throughout. Equity-Oriented Health Care Scale actually translates to an acronym of EOHCS or EoHCS, not EHoCS. I see how EHoCS is easier to pronounce as a word, but the mis-ordered letters are confusing. The episodic care scale, EEE-HC, is quite different, although it attempts to measure similar constructs in a different setting; it is also not easily pronounced. You might consider using the same stem for each scale, with a different setting indicator, e.g., xxxx-PCC for primary care clinic and xxxx-ECED for episodic care in emergency departments. Ideally, the scale name should clearly indicate it is a measure of patient experience of equitable care.

Thank you very much encouraging us to reconsider the names for both scales. We originally named the Equity-Oriented Health Care Scale and abbreviated it as EHoCS (for easy of verbalizing the name), however, we agree that this acronym may obfuscate. Our team discussed this at length, and agree that it would be useful to have a shared stem for both scales (Equity-Oriented Health Care Scale) with indicators that specify whether the scale is intended for use in settings where care is ongoing, or episodic. The revised names are reflected in the revised manuscript: 

Equity-Oriented Health Care Scales (EOHCS)– Ongoing

Equity-Oriented Health Care Scales (EOHCS)– Episodic

We realize these do not shorten to an easily-pronounced acronym, however, we feel that these names foreground the construct measured (equity-oriented health care) and the specific contexts. 

3. Page 6, line 120. If the term “structural violence” is to be used, provide a citation and explanation of meaning.

We have provided a definition of this term. Please see lines 163-165.

4. Page 8, line 178. Five cognitive interviews seems to be a fairly low number for scale development. Why was this number deemed to be adequate? Was saturation achieved this quickly?

We have provided an explanation in this section of the paper (see lines 224-230). Specifically, we have stated that, given the precarity of peoples’ living situations (reliance on shelters, living on or near the street, struggling with food security, etc.), we were pleased to have obtained input from five individuals who had extensive experiences in the primary care sector. They were in agreement with regard to the clarity and relevance of items, and were uniformly positive about the prospect of patients being invited to respond to these items. 

5. Page 11, Table 2. The question stem does not align with some of the items, specifically those that repeat “how often…” from the question stem.

We have removed the stem, and each item has been included in full. Please see in line 298.

6. Page 12, line 251. Please provide wording for all values of the response options, including those between never (0) and always (4).

We have added additional wording for all values. Please see lines 306-307.

7. Page 12, lines 259-262. As worded here, the narrative seems to imply that path analysis produces causal inferences, which it does not. Such analysis may support predictive validity statistically and is certainly useful to general hypotheses of causality to be further researched. However, it does not indicate that the scale is a “measure” of benefits.

Thank you for pointing to the need for clarification. We have revised this sentence to say that the scale may be useful in assessing the possible impacts of interventions. Please see edits in lines 324-326.

8. Provide the final version of each scale, including all question stems, items, response options, and scoring.

The final versions of each scale have been included as supporting information. Please see S1 Table: Equity-Oriented Health Care Scale - Ongoing and S2 Table: Equity-Oriented Health Care Scale – Episodic.

---

## [Decision Letter · Decision Letter 1]

13 May 2024

Using a health equity lens to measure patient experiences of care in diverse health care settings

PONE-D-23-43135R1

Dear Dr. Browne,

We’re pleased to inform you that your manuscript has been judged scientifically suitable for publication and will be formally accepted for publication once it meets all outstanding technical requirements.

Kind regards,

Masoud Behzadifar

Academic Editor

PLOS ONE

Additional Editor Comments (optional):

Reviewers' comments:

Reviewer's Responses to Questions

**Comments to the Author**

1. If the authors have adequately addressed your comments raised in a previous round of review and you feel that this manuscript is now acceptable for publication, you may indicate that here to bypass the “Comments to the Author” section, enter your conflict of interest statement in the “Confidential to Editor” section, and submit your "Accept" recommendation.

Reviewer #1: All comments have been addressed

2. Is the manuscript technically sound, and do the data support the conclusions?

Reviewer #1: Yes

3. Has the statistical analysis been performed appropriately and rigorously? 

Reviewer #1: N/A

4. Have the authors made all data underlying the findings in their manuscript fully available?

Reviewer #1: Yes

5. Is the manuscript presented in an intelligible fashion and written in standard English?

Reviewer #1: Yes

6. Review Comments to the Author

Reviewer #1: All comments have been adequately addressed. I believe many healthcare organisations will find the scales useful in assessing their progress towards providing equity-oriented care.

7. PLOS authors have the option to publish the peer review history of their article (what does this mean?). If published, this will include your full peer review and any attached files.

Reviewer #1: **Yes: **Ibukun Abejirinde

---

## [Editor Report · Acceptance letter]

27 May 2024

PONE-D-23-43135R1 

PLOS ONE

Dear Dr. Browne, 

I'm pleased to inform you that your manuscript has been deemed suitable for publication in PLOS ONE. Congratulations! Your manuscript is now being handed over to our production team.

Kind regards, 

on behalf of

Dr. Masoud Behzadifar 

Academic Editor

PLOS ONE